Rumen bacterial community profile and fermentation in Barki sheep fed olive cake and date palm byproducts

http://orcid.org/0000-0003-4515-1075 Rabee Alaa Emara 1 Rabee_a_m@yahoo.com
Kewan Khalid Z. 1
http://orcid.org/0000-0002-7239-8033 Sabra Ebrahim A. 2
El Shaer Hassan M. 1
Lamara Mebarek 3
1 Animal and Poultry Nutrition Department, Desert Research Center , Matariya, Cairo , Egypt
2 Genetic Engineering and Biotechnology Research Institute, University of Sadat City , Sadate City, Menoufia , Egypt
3 Forest Research Institute, University of Quebec in Abitibi-Temiscamingue , Rouyn-Noranda , Canada
Franco Bernardo
Electronic publication date: 2021 Nov 17
Publication date: 2021
Volume: 9
Electronic Location ID: e12447
Received 2021 Sep 2; Accepted 2021 Oct 18
Copyright: © 2021 Rabee et al.
Copyright year: 2021
Copyright holder: Rabee et al.
License: This is an open access article distributed under the terms of the Creative Commons Attribution License, which permits unrestricted use, distribution, reproduction and adaptation in any medium and for any purpose provided that it is properly attributed. For attribution, the original author(s), title, publication source (PeerJ) and either DOI or URL of the article must be cited.
License URL: https://creativecommons.org/licenses/by/4.0/

Keywords: Barki sheep, Rumen, Bacteria, Illumina Mi Seq, Olive cake, Date palm byproducts

Funding: The authors received no funding for this work.

==============================
Rumen bacteria make the greatest contribution to rumen fermentation that enables the host animal to utilize the ingested feeds. Agro-industrial byproducts (AIP) such as olive cake (OC) and date palm byproducts (discarded dates (DD), and date palm fronds (DPF)) represent a practical solution to the deficiency in common feed resources. In this study, thirty-six growing Barki lambs were divided into three groups to evaluate the effect of untraditional diets including the AIP on the growth performance. Subsequently, nine adult Barki rams were used to evaluate the effect of experimental diets on rumen fermentation and rumen bacteria. Three rations were used: common concentrate mixture (S1), common untraditional concentrate mixture including OC and DD (S2), and the same concentrate mixture in S2 supplemented with roughage as DPF enriched with 15% molasses (S3). The animals in S2 group showed higher dry matter intake (DMI) and lower relative growth rate (RGR) as compared to the animals in S1 group. However, the animals in S3 group were the lowest in DMI but achieved RGR by about 87.6% of that in the S1 group. Rumen pH, acetic and butyric acids were more prevalent in animals of S3 group and rumen ammonia (NH3-N), total volatile fatty acids (TVFA), propionic acid were higher in S1. Rumen enzymes activities were higher in S1 group followed by S3 and S2. The bacterial population was more prevalent in S1 and microbial diversity was higher in the S3 group. Principal coordinate analysis revealed clusters associated with diet type and the relative abundance of bacteria varied between sheep groups. The bacterial community was dominated by phylum Bacteroidetes and Firmicutes; whereas, Prevotella, Ruminococcus, and Butyrivibrio were the dominant genera. Results indicate that diet S3 supplemented by OC, DD, and DPF could replace the conventional feed mixture.

Introduction

The dramatic increase in animal feed prices encouraged nutritionists to search for cheaper alternatives to the traditional feedstuffs (Al-Dabeeb, 2005). Locally available byproducts such as agricultural and industrial byproducts could be a suitable solution to reduce the cost of animal feeding (Estaún et al., 2014) as the ruminant animals can utilize the high-fiber diets (Alnaimy et al., 2017). Palm date and olive trees are grown mainly in hot countries in the Middle East and Africa where Egypt is the main producer of date palm worldwide (Almitairy et al., 2011; Awawdeh & Obeidat, 2013; Khattab & Tawab, 2018). Olive fruit processing for oil production generates large quantities of olive cake (OC) that has a negative environmental impact; therefore, many studies proved its potential as a part of animal diets (García-Rodríguez et al., 2020). On the other hand, various types of residues are produced from date palm trees such as discarded dates (DD) that are generated from dates processing and date palm fronds (DPF) that are generated from tree pruning (Almitairy et al., 2011; Allaoui et al., 2018).

DD are characterized by high energy content and palatability for animals; therefore, it is a suitable alternative for conventional cereals in animal diets (Almitairy et al., 2011). However, both OC and DPF are considered as fibrous materials, and low in protein content and high in phenolic compounds like tannins that could depress rumen fermentation and modulate rumen microbiota (Mahgoub et al., 2007; Mioč et al., 2007; Khattab & Tawab, 2018; Mannelli et al., 2018; García-Rodríguez et al., 2020). Different studies indicated that those byproducts might replace a part of the concentrates feed mixture (Al-Dabeeb, 2005; Estaún et al., 2014; Khattab & Tawab, 2018). Nevertheless, limited information regarding their effect on rumen microbiota is available (Mannelli et al., 2018; García-Rodríguez et al., 2020).

The rumen is inhabited by various microbial communities, including bacteria, fungi, protozoa, archaea, and viruses, which enable the ruminant animal to utilize the ingested feed (Gilbert et al., 2020). Bacteria dominate the rumen microbial communities and make the greatest contribution to rumen fermentation (Kim, Morrison & Yu, 2015). It can ferment a wide range of substrates in animal diet including, cellulose, xylan, amylose, and protein and produce volatile fatty acids and microbial protein that provide the host animal with a large proportion of daily protein and energy requirements (Henderson et al., 2015; Rabee et al., 2020b). The composition and density of rumen bacteria are highly influenced by diet composition (Henderson et al., 2015) as the diet modulates the rumen fermentation patterns and pH (Carberry et al., 2014). Subsequently, diets with high-cellulose and hemicellulose stimulate the fibrolytic bacteria, while concentrates feed mixture rich in starch and soluble sugars stimulate amylolytic bacteria (Carberry et al., 2012). Therefore, improving strategies to understanding the rumen microbiome could maximize animal productivity (Lee et al., 2012; Rabee et al., 2020b).

Analysis of rumen microbiome using high throughput sequencing technologies has expanded our understanding of rumen microbial communities (Jami, White & Mizrahi, 2014). The adoption of these techniques helped to explore the variation in microbial communities due to changing the animal diet or geographic location (Henderson et al., 2015). However, the final output could be biased due to the DNA extraction method, PCR-primer, sequencing platform, and bioinformatics pipeline (Henderson et al., 2013; Rabee et al., 2020b). Therefore, it is highly recommended to link rumen microbiota, functional genes, metabolic pathway, rumen metabolites, and animal performance (Denman, Morgavi & McSweeney, 2018; Li et al., 2019).

The Barki sheep is a small breed that has white wool and a brown neck and is distributed in the desert in the Mediterranean zone; also, it has long thin legs that allow travel for long distances in search of grass (Elshazly & Youngs, 2019). This breed is well adapted to survive in harsh conditions, including poor feeding, heat stress, and diseases (Abdel-Moneim et al., 2009). The Barki sheep is the main sheep breed in Egypt’s desert; it provides meat and milk under desert conditions; thus, it represents an important food supply in Egypt (Abousoliman et al., 2020). Despite the economic importance of Bakri sheep in Egypt, the rumen microbiome did not receive attention in comparison to other domesticated ruminants. Ruminant production in the Mediterranean countries is challenged by poor quality and scarcity of pastures especially during the drought periods besides the increase in cereal prices (Elshazly & Youngs, 2019). Therefore, it is important to develop new strategies to improve animal efficiency using locally available resources (Romero-Huelva, Ramos-Morales & Molina-Alcaide, 2012). Feeding systems that depend on OC, DD, and DPF are common in arid countries including Egypt; besides, several studies interested in the effect of inclusion of olive cake and date palm byproducts on animal performance and rumen fermentation (Estaún et al., 2014; Khattab & Tawab, 2018). However, there is still limited information on the impact of these residues on the rumen microbiome. Mannelli et al. (2018) incorporated olive cake into the diet of lactating ewes and observed that the relative abundances of some bacterial taxa have been affected. On the other hand, no data were available on the effect of date palm byproducts on rumen bacteria. Additionally, no previous reports studied the effect of rations including OC, DD, and DPF on animal performance, rumen fermentation, and rumen bacteria. Therefore, the objective of this study was to evaluate the effect of common untraditional feeding systems that based on OC, DD, and DPF on the growth performance, rumen fermentation, and rumen bacteria community of Barki sheep.

Materials & methods

Growth trial

Growth trial was conducted at a commercial private farm in Matrouh government as a part of project “The Executive Project for Breeding and Production Systems Development of Camel and Small Ruminant in the Northwest Coast of the Arab Republic of Egypt” that was funded by the Arab Center for Studies of Arid Zones and Arid Lands (ACSAD). The project was conducted and supervised by the Desert Research center, Egypt. Thirty-six growing male Barki lambs with an average body weight of 29.65 ± 0.29 kg (mean ± SE) were randomly divided into equal three groups and housed in 12 well-ventilated shaded pens as three replicates in each group (three lambs/pen). The animals were assigned to one of three treatments and received ad lib diets for 75 days and free access to drinking water was provided. The experimental rations were: control ration or common concentrate mixture (S1); non-traditional concentrate mixture including 10% olive cake (OC) and 60% discarded date palm (DD) (S2); the same concentrate ration in S2 supplemented with ground date palm fronds (DPF) enriched with 15% molasses as roughage (S3). The physical composition of the experimental rations is presented in Table 1. Lambs were weighed biweekly until the end of the experiment. At the end of the growth trail, the animals were left on the farm without euthanizing.

Table 1 The components of experimental diets.

Physical composition (%) of the experimental rations.

Ingredients	Farm ration	Experimental rations	
S1	S2	S3	
Corn granis	22.5	10.0	10.0	
Wheat grains	22.5	0	0	
Soybean meal	11.0	17.0	17.0	
CFM*	44	0	0	
Olive cake	0	10.0	10.0	
Discarded dates	0	60.0	60.0	
Mineral & Vitamins	0	0.3	0.3	
Salt	0	1.0	1.0	
Lime stone	0	1.3	1.3	
Yeast	0	0.3	0.3	
Anti-fungi	0	0.1	0.1	
Total	100	100	100	
Roughage	0	0	+	
R:C ratio	0/100	0/100	25/75	
Note:

* CFM (14% CP and 65% TDN): concentrate feed mixture consisted of corn 55%, un-decorticated sunflower seed meal 12%, soybean meal 10%, wheat bran 17%, vinas 3%, lime stone 1.5%, salt 1%, premix 0.5%.

Digestibility and rumen fermentation trials

Nutrients digestibility and rumen fermentation trials were conducted at Maryout Research station, Desert Research center, Egypt. Nine adult Barki rams with an average body weight of 55.10 ± 1.47 kg (Mean ± SE) were assigned into three groups (three animals/treatment) to evaluate the impact of experimental diets on the digestibility of DM, CP, and NDF as well as investigation of rumen fermentation and composition of rumen bacterial community.

Animals were adapted for the diet in individual metabolic cages for 15 days, followed by 7 days as a collection period. Drinking water was offered free before the collection period and two times during the collection period. Daily collected feces were subsampled and dried at 70 °C for 48 h, then grounded and conserved until analysis. On day 22, rumen liquid was collected using a stomach tube before the morning feeding and the pH of rumen fluid was immediately recorded using a digital pH meter (WPA CD70). Rumen samples were used for the analyses of rumen ammonia and VFA, DNA extraction, and lignocellulolytic enzymes assays.

This study, including the growth trail and digestibility trail, was conducted under guidelines set by the Department of Animal and Poultry Production, Desert Research Center, Egypt. Moreover, the project was approved by the Institutional Animal Care and Use Committee, Faculty of Veterinary Medicine, University of Sadat City, Egypt (Reference: VUSC00008). All methods were performed in compliance with the ARRIVE guidelines. In addition, the project does not include euthanasia of the experimental animals. The sample size was decided based on the availability of animals that are similar in age, weight, physiological stage.

Chemical analyses

The experimental diets were analyzed for dry matter, crude protein (CP), and crude fiber (CF) according to AOAC (1997). Neutral detergent fiber (NDF) and acid detergent fiber ADF contents were determined by the method of Van Soest, Robertson & Lewis (1991) without sodium sulfite. Dry fecal samples were analyzed for CP and NDF. Moreover, the rumen ammonia and total VFA concentrations were determined by steam distillation in a Kjeldahl distillation according to the methods of Annison (1954) and AOAC (1997), respectively. In addition, individual VFAs were measured using high-performance liquid chromatography (HPLC) (Weimer, Shi & Odt, 1991). Additionally, cellulase and xylanase activities were determined by quantifying the released reducing sugars by the 3, 5-dinitrosalicylic acid (DNS) (Ghose, 1987; Bailey, Biely & Poutanen, 1992). Xylanase was measured as endo-xylanase that was defined as the amount of enzyme that releases 1 μmol of xylose per ml in a minute. Cellulase was quantified as a unit of endo-β-1,4-glucanase that is defined as the amount of enzyme that could hydrolyze filter paper and release 1 µmol of glucose within 1 min of reaction.

DNA extraction, PCR amplification, and sequencing

One milliliter of rumen fluid was centrifuged at 13,000 rpm and the precipitated pellets were used for DNA extraction by i-genomic Stool DNA Extraction Mini Kit (iNtRON Biotechnology, Inc., Seongnam-si, South Korea) according to the manufacturer’s instructions. DNA was then eluted in 50 µL elution buffer and DNA quality and quantity were verified using agar gel electrophoresis and Nanodrop spectrophotometer (Thermo Fisher Scientific, Madison, WI, USA). DNA Amplicon libraries targeting the V4–V5 region of the 16S rRNA bacterial 16S ribosomal DNA gene were generated by PCR amplification using primers 515F (5′-GTGYCAGCMGCCGCGGTAA-3′) and 926R (5′-CCGYCAATTYMTTTRAGTTT-3′) (Walters et al., 2015). PCR amplification was conducted under the following conditions: 94 °C for 3 min; 35 cycles of 94 °C for 45 s, 50 °C for 60 s, and 72 °C for 90 s; and 72 °C for 10 min. PCR products’ purification, preparation for sequencing using Illumina MiSeq system were conducted according to the protocol described by Comeau, Douglas & Langille (2017) in Integrated Microbiome Resource (Dalhousie University, Canada).

Quantitative real-time PCR

Real-time PCR was conducted to determine the total bacterial 16S rRNA copy number in the rumen fluid. Standards were generated using dilutions of purified genomic DNA from Prevotella sp, Ruminococcus albus, Butyrivibrio hungatei purchased from DSMZ (Braunschweig, Germany). Dilution series of the standards ranging from 101 to 106 copies of the 16S rRNA gene were used. The qPCR was performed using the Applied Biosystems StepOne system (Applied Biosystems, Foster City, CA, USA).

The bacterial specific primers F (5′-CGGCAACGAGCGCAACCC-3′) and R (5′-CCATTGTAGCACGTGTGTAGCC-3′) (Denman & McSweeney, 2006) were applied to amplify DNA samples and diluted standards. The 10-μL reaction consisted of 1 μL genomic DNA, 1 μL of each primer, and 7 μL SYBER Green qPCR- master mix (iNtRON Biotechnology, Inc., Seongnam-si, South Korea). The PCR conditions were as follows: 40 cycles of 95 °C for 15 s, and 60 °C for 60 s. The linear relationship between the threshold amplification (Ct) and the logarithm of 16S rDNA copy numbers of the standards was used to calculate the copy numbers of rumen bacteria per μL of DNA.

Bioinformatics analyses

The bioinformatics analyses of the paired-end (PE) Illumina raw sequences were processed in R (version 3.5.2) using DADA2 (version 1.11.3) (Callahan et al., 2016). Briefly, reads will be denoised, dereplicated and filtered for chimeras to generate Amplicon Sequence Variants (ASVs). Taxonomic assignment of sequence variants was compared using the latest SILVA reference database SILVA. The resulting ASV table was normalized and subsequently used to perform downstream analyses, including the computing of alpha and beta diversity metrics and taxonomic summaries.

Statistical analysis

The statistical analyses were conducted using the IBM SPSS version 20 software (SPSS, 1999). The differences in feed intake, relative growth rate, rumen fermentation parameters, rumen enzymes, bacterial copy number, microbial diversity, and relative abundance of bacterial phyla and genera were performed using one-way ANOVA based on a post hoc Duncan test. For all statistical tests, p-values < 0.05 were considered significant. The results of relative abundance of bacteria were tested for normality using Shapiro–Wilk test and non-normal values were then arcsine transformed. All the sequences were deposited to the sequence read archive (SRA) under the accession number: PRJNA744569.

Results

Chemical Composition of the experimental diets

The composition and chemical analysis of animal diets are presented in Tables 1 and 2. The results indicated that olive cake and discarded date palm represented 70% of the untraditional concentrates mixture (S2). Moreover, the traditional concentrates mixture (S1) contained higher content of CP and NDF compared with S2. Whereas, S2 contained higher content of DM, CF, and ADF compared with S1.

Table 2 Chemical analysis of experimental diets.

Chemical composition of concentrate feed mixtures and date palm frond.

Items	Concentrate feed mixture	Date palm fronds	
S1	S2	
DM (g/kg)	924.1	954.5	986.7	
CF (g/kg DM)	48.0	95.1	298.4	
CP (g/kg DM)	200.0	185.2	75.3	
NDF (g/kg DM)	551.0	368.8	538.9	
ADF (g/kg DM)	85.3	184.7	359.8	

Performance of growing lambs

The results of dry matter intake (DMI), organic matter intake (OMI), and the relative growth rate (RGR) are presented in Table 3. The differences in DMI and RGR were significant among groups. Animal group S2 consumed the highest total DMI followed by S1 and S3, respectively. Both S2 and S3 groups showed a comparable RGR value but they are lower than that revealed in the S1 group.

Table 3 Growth performance and digestibility.

Effect of diet type on growth performance of growing lams and nutrients digestibility of the experimental rations.

Items	Feeding systems	SEM	Mean	P value	
S1	S2	S3	
Growth experiment	
Lamb No.	12	12	12	–		–	
IBW, kg	29.20	29.63	29.97	0.51	29.5	0.540	
RGR*, %	66.02	54.43	57.78	1.02	59.4	0.001	
Dry CFMI, g DM/kg0.75	106.9	112	70.68	0.38	96.45	0.001	
Dry RghI, g DM/kg0.75	0	0	23	–		–	
TDMI g/kg0.75	106.9	112	93.67	0.44	104.2	0.001	
Digestibility experiment	
Rams No.	3	3	3	–		–	
BW, kg	42.03	42.7	42.58	0.31	42.3	0.314	
Dry CFMI/kg0.75	61.91	66.02	46.69	0.73	58.2	0.001	
Dry RghI/kg0.75	0	0	16.01	0.38		–	
TDMI/kg0.75	61.91	66.02	62.70	0.91	63.5	0.041	
DM digestibility, %	81.87	67.56	67.73	1.99	72.4	0.003	
CP digestibility, %	73.21	63.70	58.64	2.82	65.2	0.028	
NDF digestibility, %	86.58	60.74	60.17	2.26	69.15	0.001	
Note:

* Relative Growth Rate (RGR), % = (final BW – initial BW) × 100/IBW, BW = Body Weight, CFMI = Concentrates feed mixture intake, RghI = Roughage intake, TDMI = Total dry matter intake, SEM = Standard Error Mean.

Digestibility of nutrients in adult rams

Sheep group S2 showed the highest total DMI followed by S3 and S1, respectively (Table 3). The untraditional diets (S2 and S3) showed lower values for CP and NDF digestibility as compared to the traditional diet (S1).

Rumen fermentation parameters and lignocellulolytic enzymes

The effect of ration type on rumen fermentation parameters and enzymes assays are presented in Table 4. The ration S3 that containing DPF resulted in a higher rumen pH value followed by S2 and then S1; and rumen ammonia concentration followed the same trend. Both S2 and S3 showed significantly lower total VFA concentration but higher in acetic and butyric fractions as compared with S1. However, the ration S1 was superior in propionic acid fraction.

Table 4 Rumen fermentation parameters and enzymes activites.

Rumen pH, ammonia nitrogen (NH3-N), total volatile fatty acid (VFA) concentration and VFA proportions, cellulase and xylanase enzymes, and bacterial population (Log10 copies/μL DNA) in the rumen of Barki sheep under investigation (Mean ± SE).

Item	Feeding systems	SEM	Mean	P value	
S1	S2	S3	
Animal numbers	3	3	3	9	9		
pH	5.5	6.2	6.5	0.15	6.04	0.0001	
NH3-N, mg/dl	10.94	4.5	2.2	1.13	5.85	0.0001	
TVFA, meq/dl	8.2	5.4	5.5	0.5	6.4	0.006	
Acetic acid, %	34.7	50.7	54.7	3.8	46.7	0.006	
Propionic acid, %	62.0	44.3	36.5	4.0	47.6	0.002	
Butyric acid, %	3.3	5.0	8.8	0.8	5.7	0.0001	
A/P ratio	0.56	1.14	1.50	–	–	–	
Xylanase, IU/ml	9.8	8.3	3.8	1.8	7.3	0.441	
Cellulase, IU/ml	8.7	6.7	8.3	1.0	7.9	0.731	
BP*, Log10 copies/μL DNA	5.65	3.1	4.4	0.40	4.4	0.029	
Note:

* Bacterial population.

S1 group showed higher values for rumen xylanase and cellulase production as compared with the other two groups but still without significant differences (Table 4). The bacterial population (Log10 of 16S rDNA copies) was significantly affected by the experimental rations. The copy number was higher in the S1 group followed by S3 and then S2, respectively (Table 4).

Rumen bacterial diversity analysis

Following quality filtering, merging and removal of chimeric sequences, the sequencing of the V4 region on 16S rDNA in rumen samples from nine sheep resulted in 196999 high-quality sequence reads with an average of 21,888 ± 1,113 reads per animal (Mean ± Standard error; SE) (Table 5). Alpha diversity metrics were used to estimate the similarity in the bacterial community among sheep groups. The bacterial richness was estimated using Chao and ACE indices. Moreover, the bacterial diversity was determined using Invsimpon and Shannon indices. The sheep group fed S3 diet showed higher ASVs number and alpha diversity indices compared to S2 and S1 groups with a significant difference in Shannon index. The principal coordinate analysis (PCoA) based on Bray-Curtis dissimilarity (Fig. 1) used to estimate beta diversity of bacterial communities within sheep groups confirmed that sheep groups were separated distinctly from each other.

Figure 1 Principal coordinates analysis (PCoA).

Principal coordinates analysis of Barki sheep rumen bacterial community based on Bray-Curtis dissimilarity. The analysis was performed between three sheep: black circles for samples of group S1, red triangles for samples of group S2, and green squares for samples of group S3.

Table 5 Alpha diversity indices.

Summary of ASVs number, bacterial diversity indices in the rumen of Barki sheep under investigation (Mean ± SE).

Diversity indices	Feeding systems	SEM	Mean	P value	
S1	S2	S3	
Animal numbers	3	3	3	9	9	–	
Sequence reads*	21,023	21,785	22,857	1,113	21,888	0.838	
ASVs	1,517	2,146	2,479	242	2,047	0.292	
Chao1	1,519	2,147	2,481	242	2,049	0.292	
ACE	1,526	2,157	2,498	244	2,060	0.289	
Shannon	5.34	6.85	7.07	0.3	6.4	0.008	
Invsimpson	112	458	657	105	409	0.081	
Fisher	254.8	471.5	572.3	63.5	432.9	0.095	
Note:

* Average/sample.

The composition and relative abundance of bacterial community

A total of 14 bacterial phyla were detected and the bacterial community was dominated by phylum Bacteroidetes (76.2%) and Firmicutes (19.9%) and other phyla found to be less than 1.5% (Table 6). The relative abundance of bacterial phyla varied between sheep groups. Phylum Bacteroidetes dominated the bacterial community in sheep rumen and it was higher in the S3 (78.7%) group compared to S1 (75.8%) and S2 (74.1%) (Table 6 and Fig. 2). This phylum was dominated by four families, including Rikenellaceae, and Prevotellaceae. Family Rikenellaceae showed the highest proportion in the S2 group and was dominated by RC9_ gut group. In addition, family Prevotellaceae was higher in S1 group with a significant difference and most of the family’s reads were assigned to genus Prevotella and uncultured Prevotellaceae.

Figure 2 The relative abundance of dominant bacterial phyla.

The relative abundance of dominant bacterial phyla in the rumen of three sheep groups (S1, S2, S3).

Table 6 Bacterial phyla in the sheep rumen.

Relative abundance of bacterial phyla in the rumen of Barki Sheep under investigation.

Bacterial phyla	Feeding system	SEM	Mean	P value	
S1	S2	S3	
Animal number	3	3	3		9		
Bacteroidetes	75.8	74.1	78.7	1.2	76.2	0.318	
Firmicutes	22.7	22.1	15.0	1.5	19.9	0.027	
Proteobacteria	0.05	0.8	1.03	0.24	0.6	0.262	
Spirochaetes	1.02	0.8	0.8	0.17	0.86	0.867	
Actinobacteria	0.34	0.13	0	ND	ND	ND	
Fibrobacteres	0.07	0.04	0.4	0.07	0.18	0.003	
Planctomycetes	0.06	0.8	1.15	0.2	0.76	0.041	
Tenericutes	0	0.9	1.47	ND	ND	ND	
Cloacimonetes	0	0.046 (1n)	0.32	ND	ND	ND	
Elusimicrobia	0	0.12	0.37	ND	ND	ND	
Kiritimatiellaeota	0	0.09	0.36	ND	ND	ND	
Verrucomicrobia	0	0.05 (1n)	0.17	ND	ND	ND	
Synergistetes	0.05	0.1	0.18	0.03	0.12	0.121	
Lentisphaerae	0	0	0.18	ND	ND	ND	
Note:

ND, Non-determined; 1n, the phylum was observed in one sample.

Phylum Firmicutes was the second largest phylum in sheep rumen, it was significantly higher in the S1 group (22.7%), followed by S2 (22.1%), and S3 (15%) (Table 6 and Fig. 2). This phylum was dominated by five families, including Veillonellaceae, Erysipelotrichaceae, Ruminococcaceae, Lachnospiraceae, and Family_XIII. While group S2 showed the highest representation of family Lachnospiracea, the sheep group S1 diet showed the highest representation of Veillonellaceae, Erysipelotrichaceae, and Ruminococcaceae. Furthermore, Family_XIII was higher in S3. In addition, phylum Firmicutes was dominated by five genera, including Selenomonas, Ruminococcus, Butyrivibrio, Megasphaera, and Sharpea. These genera were also affected by diet type; Selenomonas was higher in S1group followed by S3 and S2, respectively. Butyrivibrio was higher in S2 followed by S3 and S1 respectively (Table 6 and Fig. 2).

Sheep in the S3 group showed higher proportions of phylum Proteobacteria, Fibrobacteres, Planctomycetes, Tenericutes, Cloacimonetes, Elusimicrobia, Kiritimatiellaeota, Verrucomicrobia, Synergistetes, and Lentisphaerae. Whereas group S1 showed higher proportions of phylum Spirochaetes, and Actinobacteria and the differences between the groups were significant (P < 0.05) in the relative abundance of Fibrobacteres, and Elusimicrobia (Table 6).

Some genera were observed exclusively in specific groups such as Bifidobacterium (Actinobacteria) and Lactobacillus (Firmicutes), Schwartzia (Firmicutes), Dialister (Firmicutes), and Catenisphaera (Firmicutes) that were observed in S1. Additionally, genus Papillibacter (Firmicutes), found only in R3 and Kandleria (Firmicutes) found only in the S2 group.

Discussion

Chemical composition of diets

Physical and chemical characteristics of animal diet are the main driver of modulations in the rumen microbiome (Denman, Morgavi & McSweeney, 2018); therefore, understanding the rumen microbiome could lead to developing strategies to improve animal productivity (Wirth et al., 2018). Agricultural byproducts provide suitable alternatives to the common feedstuffs in the arid countries (Mioč et al., 2007; Awawdeh & Obeidat, 2013; Allaoui et al., 2018). However, those byproducts should be evaluated on the level of their effect on animal performance and the rumen ecosystem (Alnaimy et al., 2017). To the best of our knowledge, this study is the first to report the effect of using untraditional diets that contain a combination of olive cake and date palm byproducts (discarded date or dates palm fronds) in sheep feeding on animal performance, rumen fermentation, and rumen bacteria (Table 1). The common concentrate feed mixture is a starchy mixture and has low-fiber content and high content of crude protein, energy, and soluble carbohydrates (Carberry et al., 2012). At the same time, olive cake and date palm fronds are described as low-quality feedstuffs with high contents of lignocelluloses and low contents of protein and energy (Awawdeh & Obeidat, 2013; Boufennara et al., 2016; García-Rodríguez et al., 2020). Furthermore, the discarded dates are considered as an energy source due to high soluble carbohydrates and contain low crude protein (Al-Dabeeb, 2005; Boufennara et al., 2016; Allaoui et al., 2018).

Animals’ performance

In this study, the highest DMI was obtained by group S2, which fed feed mixture contains OC, and DD; while the lowest DMI was obtained by group S3, which fed feed mixture contains OC, and DD plus DPF (Table 3), this finding in line with a study by Rajabi et al. (2017), when discarded dates included in sheep’s diet. De Visser et al. (1998) indicated that the inclusion of the late cut grass with high-fiber content decreased the DMI. The higher DMI in S2 could be attributed to the palatability of diet contains DD or rapid passage rate (Khezri, Dayani & Tahmasbi, 2017; Iqbal et al., 2019).

Untraditional CFM diet alone (S2) or supplemented with date palm fronds (S3) resulted in low relative growth rate (54.43 vs. 57.78%, respectively) but the highest value was recorded for the S1 group (66.02%) that fed common feed mixture (Table 3). The decline in growth rate in S2 and S3 diets could be attributed to low protein and high fiber content compared to the S1 diet (Kewan, 2013; Allaoui et al., 2018). This finding is similar to previous studies that indicated that the inclusion of the olive cake or discarded dates declined the growth performance (Al-Dabeeb, 2005; Mioč et al., 2007; Allaoui et al., 2018). In addition, Abo Omar, Daya & Ghaleb, 2012 mentioned that the inclusion of olive cake into Awassi lambs’ diet did not affect the performance adversely. Mahgoub et al. (2007), reported that sheep fed date palm fronds gained less weight than those fed rhodes grass hay; which could be due to higher fiber content and lower protein content in palm fronds (Khattab & Tawab, 2018).

The decline in growth rate in S2 and S3 diets groups could be attributed to lower dietary protein and high fiber contents compared to the S1 diet (Al-Dabeeb, 2005; Allaoui et al., 2018). More studies mentioned that incorporation the olive cake and date palm byproducts in lamb diets declined the digestibility of DM, OM, CP, NFE, and CF (Al-Dabeeb, 2005; García-Rodríguez et al., 2020), which support our results and could be attributed to high fiber and phenolic compounds that limit the availability of nutrients and affect the rumen microbiota negatively (Mioč et al., 2007; Awawdeh & Obeidat, 2013; Boufennara et al., 2016; Djamila & Rabah, 2016). Therefore, animals fed on such byproducts will require protein supplementation (Allaoui et al., 2018); also, feeding the animals on crude byproducts should be avoided (Mioč et al., 2007).

Lower digestibility of CP for both S2 and S3 than S1 (Table 3) might be due to higher condensed tannins in dates, which bind with protein and reduced its utilization. These results are similar to the findings that reported by Kewan (2013) who found that CP digestibility was 71.49 and 77.38 for lambs fed concentrate feed mixture with complete replacement of corn grains (27.5% of the TMR) by discarded dates and control group, respectively. In addition, lower digestibility of NDF for both S2 and S3 than S1 might be due to the soluble carbohydrate in the dates which has commonly been associated with rapid fermentation and subsequent depression of ruminal pH. This finding was confirmed by Khattab, Abdel-Wahed & Kewan (2013) who found a linear decrease of NDF digestibly by increasing the dates inclusion rate.

Rumen fermentation activity

The improvement in rumen pH in S2 and S3 compared to S1 diet (Table 4) was also indicated in other studies on olive cake (Awawdeh & Obeidat, 2013; García-Rodríguez et al., 2020). In addition, the incorporation of the DPF in the S3 diet has increased the rumen pH compared to the other two groups. Khattab & Tawab (2018) reported that the inclusion of DPF in sheep’s diet kept the rumen pH higher than sheep fed clover hay. The decline in rumen pH has a negative consequences on rumen fermentation (Dijkstra et al., 2012), which could be a positive point for using S3 diets in sheep feeding. The concentrate feed mixtures consist of rapidly fermentable carbohydrate that encourages the VFA production that declines the rumen pH; in contrast, the higher content of ADF and NDF in S3 diets has neutralized the rumen pH (Asadollahi et al., 2016). The higher concentration of ammonia in S1 is reflected by the higher dietary protein compared to other diets (Hamchara et al., 2018; Khattab & Tawab, 2018; García-Rodríguez et al., 2020). Previous studies showed that incorporation of the DD in the animal diet decreased the protein content, rumen ammonia, and VFA; and did not affect VFA composition (Rajabi et al., 2017; Khezri, Dayani & Tahmasbi, 2017). Another explanation for the linear decrease of ammonia is soluble sugars in the S2 and S3 capture more degradable nitrogen for microbial protein synthesis than the starch in S1 (Khezri, Dayani & Tahmasbi, 2017).

The inclusion of OC, DD, and DPF in the diet changed the proportions of individual VFAs. Acetic and butyric acids were increased a long with a decrease in propionic acid (Table 4); this finding is in agreement with previous studies (De Visser et al., 1998; Pallara et al., 2014; Asadollahi et al., 2016; García-Rodríguez et al., 2020) that revealed that higher dietary fiber stimulates the production of acetic acid and decrease the propionic acid due to increasing the activity of fibrolytic bacteria. The increase of butyrate in S2 and S3 groups was also obtained in previous studies on DD in sheep diets (Khezri, Dayani & Tahmasbi, 2017). Analyzing the effect of the feeding system on VFA production in the rumen is important as VFAs provide the animal with two-third of the energy supply of the ruminant (De Visser et al., 1998). Consequently, the production of meat and milk could be affected; for instance, the reduction in acetic to propionic ratio has a depressing effect on milk fat production (Sutton et al., 2003).

The activities of rumen enzymes and VFA production are the reflection of rumen microbial groups involved in rumen fermentation (Kamra, Agarwal & McAllister, 2010; Latif et al., 2014). According to a study by Raghuvansi et al. (2007), the increase in the production of fibrolytic enzymes could be attributed to the higher numbers of fibrolytic bacteria. Our results indicated that group S1 showed higher enzymes activities (cellulase and xylanase), and a higher bacterial population followed by S3 (Table 4); this finding could be attributed to the presence of rapidly degradable carbohydrates and degradable nitrogen in S1 groups that stimulated bacterial growth (Raghuvansi et al., 2007; Azizi-Shotorkhoft et al., 2018). Moreover, higher cellulase production and bacterial population in S3 compared to S2 could be explained by the presence of molasses in the diet and higher content of ADF, which agrees with the results of Azizi-Shotorkhoft et al. (2018) who indicated that the inclusion of molasses increased the cellulolytic activities. On the other hand, Khattab & Tawab (2018) reported that the inclusion of palm fronds in the animal diet reduced microbial protein due to low protein content that depresses microbial growth. The information about the effect of OC, DD, and DPF on rumen enzymes and the bacterial population is limited; an In vitro study conducted by García-Rodríguez et al. (2020) reported an increment in the bacterial density and reduction of the microbial protein by incorporation olive cake into the animal diet. Moreover, Dahlan (2000) showed that goat fed oil palm fronds produced more ammonia, TVFA, acetic acid, cellulase and xylanase compared to goat fed rice straw, which supports our findings. Sheikh et al. (2019) revealed that feed digestibility and microbial activity were improved when degradable protein was added to the diet, which could explain the higher bacterial population and VFA production in S1.

Diversity and composition of rumen bacteria

Inclusion of the OC, DD, and DPF to the concentrate mixture has changed the diversity and composition of rumen bacteria (Tables 5 and 6 and Figs. 1 and 2), which is in agreement with previous studies on the olive cake (Pallara et al., 2014; Mannelli et al., 2018; García-Rodríguez et al., 2020). Mannelli et al. (2018) reported that rumen microbiota is highly affected by dietary composition especially the chemical compounds that have antimicrobial activities such as polyphenols observed in byproducts. At the same time, no available data on the effect of discarded dates and date palm fronds on the rumen microbial populations. The higher ASVs were linked with sheep group fed S3 diet that contains concentrate mixture plus forage, which is similar to findings obtained by Petri et al. (2012) and Rabee et al. (2020a). In this study, phylum Bacteroidetes and Firmicutes dominated the bacterial community; this finding was also reported in the rumen of lactating ewes fed olive cake (Mannelli et al., 2018), cattle (Petri et al., 2013), and camel (Rabee et al., 2020a).

Firmicutes were dominated by the family Ruminococcaceae and Lachnospiraceae (Tables 6 and 7), which is in agreement with results obtained by Rabee et al. (2020a). Members of family Ruminococcaceae degrade the hemicellulose, pectin, and cellulose present in the plant cell wall (Pettipher & Latham, 1979), which could explain the high representation of this family in S1 group that fed mixture with high-NDF content and group S3 that fed mixture with high-ADF and NDF contents. The decline in the relative abundance of Ruminococcaceae in S3 compared to S1 could be attributed to the presence of phenolic compounds in olive cake and date palm byproducts that affect fibrolytic bacteria (Khattab & Tawab, 2018; Mannelli et al., 2018). According to Moumen et al. (2007), the optimal fermentation of lignocellulosic byproducts could be achieved by the addition of nitrogen and soluble carbohydrates to animal diets, which enhance rumen fermentation; this might illustrate the higher abundance of Ruminococcaceae in S1.

Table 7 Bcaterial genera in the sheep rumen.

Relative abundances of dominant bacterial families and genera in the rumen of Barki sheep under investigation (Mean ± SE).

Family	Genus	Feeding system	SEM	Mean	P value	
S1	S2	S3	
Animal number		3	3	3	9	9		
Phylum: Actinobacteria	
Bifidobacteriaceae	Bifidobacterium	0.025	0	0	ND	ND	ND	
Phylum: Bacteroidetes			
Order:Bacteroidales		75.6	74.02	78.7	1.2	76.1	0.318	
Family: Rikenellaceae		9	26.05	15.7	3.5	16.9	0.132	
Rikenellaceae	U29-B03	0.026	0.19	0.18	0.03	0.12	0.184	
NA	0.056	0.14	0.06	0.01	0.08	0.007	
RC9_gut_group	8.9	25.7	15.5	3.5	16.7	0.134	
Prevotellaceae		59.8	30.6	21.5	6.6	37.3	0.015	
Prevotellaceae	Prevotella	18.8	21.8	11.5	3.08	17.4	0.434	
Uncul_Prevotellaceae	40.98	8.8	9.6	6.4	19.8	0.031	
BS11_gut_group		0	0.19	0.88	ND	ND	ND	
Phylum: Firmicutes			
Lactobacillaceae	Lactobacillus	0.19	0	0	ND	ND	ND	
Streptococcaceae	Streptococcus	0	0.07	0.05	ND	ND	ND	
Veillonellaceae	Selenomonas	8.94	1.3	0.2	3.5	3.5	0.011	
Anaerovibrio	0.24	0.3	0	ND	ND	ND	
Schwartzia	0.48	0	0	ND	ND	ND	
Acidaminococcaceae	Succiniclasticum	0.9	0.86	0.77	0.8	0.8	0.941	
Acidaminococcus	0.23	0	0	ND	ND	ND	
Veillonellaceae	Megasphaera	0.036	3.78	0	ND	ND	ND	
Dialister	0.1	0	0	ND	ND	ND	
Erysipelotrichaceae	Asteroleplasma	0	0.023	0	ND	ND	ND	
UCG-004	0.07	0.5	2.1	0.34	0.9	0.008	
Sharpea	1.27	5.6	0	ND	ND	ND	
Kandleria	0	0.18	0	ND	ND	ND	
Catenisphaera	0.15	0	0	ND	ND	ND	
Ruminococcaceae		6.8	5.1	5.75	1.1	5.9	0.869	
Ruminococcaceae	Saccharofermentans	0	0.14	0.19	ND	ND	ND	
Papillibacter	0	0	0.3	ND	ND	ND	
Lachnospiraceae		1.9	5.2	4	0.56	3.7	0.021	
Lachnospiraceae	Acetitomaculum	0.09	0	0	ND	ND	ND	
Butyrivibrio	0.01	1.27	0.9	0.1	0.7	0.001	
Shuttleworthia	0.3	0.2	0.23	0.05	0.25	0.816	
Family_XIII		0.43	0.57	0.23	0.07	0.4	0.225	
Family_XIII	Mogibacterium	0.16	0.03	0	ND	ND	ND	
Anaerovorax	0	0.17	0.19	ND	ND	ND	
Christensenellaceae	R-7_group	0.02	1.13	1.1	0.2	0.77	0.002	
Phylum: Elusimicrobia			
Elusimicrobiaceae	Elusimicrobium	0	0.03	0.17	ND	ND	ND	
Phylum: Spirochaetes			
Spirochaetaceae Spirochaetaceae	Sphaerochaeta	0.48	0.19	0.27	0.07	0.3	0.192	
Treponema_2	1.59	0.67	0.35	0.2	0.8	0.251	
Phylum: Tenericutes			
Anaeroplasmataceae	Anaeroplasma	0.01	0.06	0.4	0.07	0.15	0.081	
Note:

ND, non-determined; SEM, standard error mean.

Family Lachnospiraceae was dominated by genus Butyrivibrio that is cellulolytic bacteria and was overrepresented in S2 and S3 groups that fed diets with higher ADF that favors the cellulolytic bacteria (Pitta et al., 2014; Liu et al., 2017). The decline in the abundance of this genus in S1 could be a result of lower fiber content (Mannelli et al., 2018). In addition, the genus Butyrivibrio is involved in the production of butyrate that explains the higher amount of butyrate in S2 and S3 (Bharanidharan et al., 2018). Otherwise, the decline of Butyrivibrio in S3 compared to S2 could be attributed to the presence of DPF that has higher polyphenol content (Boufennara et al., 2016; Mannelli et al., 2018). Anaerovibrio within family Veillonellaceae, has an essential role in the lipid metabolism and biohydrogenation in the rumen, this process is highly associated with concentrate content in animal diet that enhances the lipolysis (Prins et al., 1975). Consequently, the presence of this genus in the S1 and S2 groups is explained. Moreover, the disappearance of this genus from the S3 group could be linked to high fiber and polyphenols content due to the inclusion of date palm fronds that rich in both fiber and polyphenols (Mannelli et al., 2018; Boufennara et al., 2016). Thus, the polyphenols have an adverse effect on some of the bacterial genera involved in biohydrogenation in the rumen (Frutos et al., 2004; He, Lv & Yao, 2007). Consequently, the quality of animal products (milk and meat) could be improved by the incorporation of polyphenols in ruminant diets (Pallara et al., 2014). Vasta et al. (2010) reported that the inclusion of polyphenols in ruminant diets affects rumen metabolism, decreasing dietary protein degradation and fatty acid biohydrogenation through targeting specific groups of microorganisms.

At the phylum level, Bacteroidetes dominated the bacterial community in sheep groups, and its relative abundance varied between groups (Table 6 and Fig. 2). This phylum was dominated by uncultured bacteria that are specialized in lignocellulose degradation (Naas et al., 2014), which explains the higher abundance in the S3 group. In addition, this phylum was dominated by Prevotella and RC9_gut_group, which is in agreement with a finding of Fouts et al. (2012) on cows. Prevotella can degrade starch, hemicelluloses, pectin and could produce propionate and xylanase in the rumen (Russell & Rychlik, 2001). Consequently, the overrepresentation of this genus in the S1 group is expected and explains the higher propionic acid and xylanase enzymes in the S1 group. RC9_gut_group was previously observed in camel and Rhinoceros gut and could have a role in the degradation of lignocellulose, which might illustrate its higher representation in group S3 (Mackenzie et al., 2015; Rabee et al., 2020b).

Furthermore, phylum Fibrobacteres was more abundant in group S3 (Table 7), which is an expected trend as this phylum is the major cellulolytic bacteria in the rumen (Ransom-Jones et al., 2012; Petri et al., 2012). Phylum Elusimicrobia and Verrucomicrobia were found higher in the S3 group; some members of these phyla have a role in fiber degradation, which supports our results (Hou et al., 2008; Herlemann et al., 2009; Lee et al., 2009). The members of Proteobacteria have a role in protein degradation (Liu et al., 2017).

Conclusions

Olive cake and date palm byproducts are available feed resources that could replace common concentrate feed mixture especially in case of limited availability or high price. Inclusion of OC, DD, and DPF affected animal performance and rumen fermentation parameters as well as microbial diversity and the relative abundance of rumen bacteria. Additionally, lignocellulolytic bacteria showed an increase in their relative abundance in the rumen of sheep fed byproducts. The composition of the rumen bacterial community in Barki sheep is similar to other ruminant animals.

Supplemental Information

Supplemental Information 1 Raw data.

Individual records of rumen fermentation parameters, enzymes, bacterial population, and Relative Growth Rate

Click here for additional data file.

Supplemental Information 2 The ARRIVE guidelines 2.0: author checklist.

Click here for additional data file.

The authors are grateful to the Environmental and Food biotechnology Laboratory, Genetic Engineering and Biotechnology Research Institute- University of Sadat City funded by Science and Technology Development Fund (STDF), Egypt, for the use of their instruments.

Additional Information and Declarations

Competing Interests

Author Contributions

Animal Ethics

Data Availability

The authors declare that they have no competing interests.

Alaa Emara Rabee conceived and designed the experiments, performed the experiments, analyzed the data, prepared figures and/or tables, authored or reviewed drafts of the paper, and approved the final draft.

Khalid Z. Kewan conceived and designed the experiments, performed the experiments, prepared figures and/or tables, authored or reviewed drafts of the paper, and approved the final draft.

Ebrahim A. Sabra conceived and designed the experiments, performed the experiments, prepared figures and/or tables, authored or reviewed drafts of the paper, and approved the final draft.

Hassan M. El Shaer conceived and designed the experiments, analyzed the data, authored or reviewed drafts of the paper, and approved the final draft.

Mebarek Lamara conceived and designed the experiments, performed the experiments, analyzed the data, prepared figures and/or tables, authored or reviewed drafts of the paper, and approved the final draft.

The following information was supplied relating to ethical approvals (i.e., approving body and any reference numbers):

The Institutional Animal Care and Use Committee, Faculty of Veterinary Medicine, University of Sadat City, Egypt approved the study (Approval reference number: VUSC00008).

The following information was supplied regarding data availability:

The raw data is available at SRA: PRJNA744569 and in the Supplemental File.

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
