# Peer review of "Rumen bacterial community profile and fermentation in Barki sheep fed olive cake and date palm byproducts"

_PeerJ, doi:10.7717/peerj.12447_

## Round 0.1 · original submission · Minor Revisions

Dear authors,

I have read your manuscript along with two leading experts in the field and I conclude that the study has high scientific quality and only minimal issues are found. I kindly request that the authors address the issues found by both reviewers.

Congratulations on this excellent study.

Best regards

·

Basic reporting

the study is well done the use of the language is adequate, making the paper easy to read and to understand.

It is also well referenced and enough context to understand the study is presente din the introduction.

Tables and figures are OK and also discussion.

Experimental design

The study is well designed, the research question well defined, and it was done with enough rigor. In addition, the description of the methods provide enough details for repeating the experiments. Nevertheless I have a question:

how the composition of the feeding mixtures were defined see:

"The experimental rations were as follows:
S1: a common concentrate mixture served as a control ration; S2: a non-traditional concentrate mixture including 10% olive cake (OC) and 60% discarded date palm (DD); and S3: the same concentrate ration in S2 supplemented with roughage as ground date palm fronds (DPF) enriched
with 15% molasses."

Validity of the findings

the findings are valid and valuable, however I have a question, in the conclusions is stated that "Animals fed on such byproducts should be supplemented by degradable protein" was the addition of degradable protein evaluated in the study? if not why?

Additional comments

please review the references and be sure all scientific and gene names are in italics.
Provide more information about the characteriztics of Barki sheep and why they are important.

Reviewer 2 ·

Basic reporting

In table 1: The third column I think should have been S2 not S1. Right?
L336: please correct the reference. Awawdeh & Obeidat (2013) did not investigate dates.

Experimental design

meet the standards.

Validity of the findings

meet the standards.

---

## Round 0.2 · accepted · Accept

Dear authors,
Thank you so much for submitting the revised version of your manuscript. I also thank you for addressing the issues indicated by the reviewers and I find the manuscript ready for publication. I am certain that the readers will receive your manuscript with joy and will get many citations.

Thank you so much and congratulations for the excellent work submitted to Peer J.

Warm regards